# NEURAL SYNTHESIS OF BINAURAL SPEECH FROM MONO AUDIO

**Alexander Richard, Dejan Markovic, Israel D. Gebru, Steven Krenn, Gladstone Butler, Fernando de la Torre, Yaser Sheikh**

Facebook Reality Labs
Pittsburgh, USA
{richardalex,dejanmarkovic,idgebru,stevenkrenn,gsbutler,yaser}@fb.com

## ABSTRACT

We present a neural rendering approach for binaural sound synthesis that can produce realistic and spatially accurate binaural sound in realtime. The network takes, as input, a single-channel audio source and synthesizes, as output, two-channel binaural sound, conditioned on the relative position and orientation of the listener with respect to the source. We investigate deficiencies of the $\ell_2$-loss on raw wave-forms in a theoretical analysis and introduce an improved loss that overcomes these limitations. In an empirical evaluation, we establish that our approach is the first to generate spatially accurate waveform outputs (as measured by real recordings) and outperforms existing approaches by a considerable margin, both quantitatively and in a perceptual study. Dataset and code are available online.[1]

## 1 INTRODUCTION

The rise of artificial spaces, in augmented and virtual reality, necessitates efficient production of accurate spatialized audio. Spatial hearing (the capacity to interpret spatial clues from binaural signals), not only helps us to orient ourselves in 3D environments, it also establishes immersion in the space by providing the brain with congruous acoustic and visual input (Hendrix & Barfield, 1996). Binaural audio (left and right ear) even guides us in multi-person conversations: consider a scenario where multiple persons are speaking in a video call, making it difficult to follow the conversation. In the same situation in a *real* environment we are able to effortlessly focus on the speech from an individual (Hawley et al., 2004). Indeed, auditory sensation has primacy over even visual sensation as an input modality for scene understanding: (1) reaction times are faster for auditory stimulus compared to visual stimulus (Jose & Praveen, 2010) (2) auditory sensing provides a surround understanding of space as opposed to the directionality of visual sensation. For these reasons, the generation of accurate binaural signal is integral to full immersion in artificial spaces.

Most approaches to binaural audio generation rely on traditional digital signal processing (DSP) techniques, where each component – head related transfer function, room acoustics, ambient noise – is modeled as a linear time-invariant system (LTI) (Savioja et al., 1999; Zotkin et al., 2004; Sunder et al., 2015; Zhang et al., 2017). These linear systems are well-understood, relatively easy to model mathematically, and have been shown to produce perceptually plausible results – reasons why they are still widely used. Real acoustic propagation, however, has nonlinear wave effects that are not appropriately modeled by LTI systems. As a consequence, DSP approaches do not achieve perceptual authenticity in dynamic scenarios (Brinkmann et al., 2017), and fail to produce metrically accurate results, i.e., the generated waveform does not resemble recorded binaural audio well.

In this paper, we present an end-to-end neural synthesis approach that overcomes many of these limitations by efficiently synthesizing accurate and precise binaural audio. The end-to-end learning scheme naturally captures the linear and nonlinear effects of sound wave propagation and, being fully convolutional, is efficient to execute on commodity hardware. Our major contributions are (1) a novel binarualization model that outperforms existing state of the art, (2) an analysis of the shortcomings of the $\ell_2$-loss on raw waveforms and a novel loss mitigating these shortcomings, (3) a real-world binaural dataset captured in a non-anechoic room.

---

[1]https://github.com/facebookresearch/BinauralSpeechSynthesis

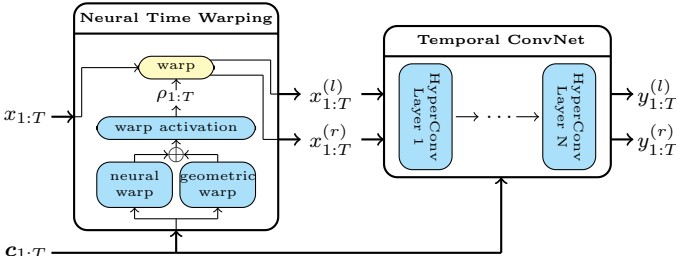

Figure 1: System Overview. Given the source and listener position and orientation $\boldsymbol{c}_{1:T}$ at each time step, a single-channel input signal $x_{1:T}$ is transformed into a binaural signal. The neural time warping module learns an accurate warp from the source position to the listeners left and right ear while respecting physical properties like monotonicity and causality. The Temporal ConvNet models nuanced effects like room reverberations or head- and ear-shape related modifications to the signal.

**Related Work.** State of the art DSP techniques approach binaural sound spatialization as a stack of acoustic components, each of which is an LTI system. As accurate wave-based simulation of room impulse responses is computationally expensive and requires detailed geometry and material information, most real-time systems rely on simplified geometrical models (Välimäki et al., 2012; Savioja & Svensson, 2015). Head-related transfer functions are measured in an anechoic chamber (Li & Peissig, 2020) and high-quality spatialization requires binaural recordings at almost 10k different spatial positions (Armstrong et al., 2018). To generate binaural audio the DSP-based binaural renderers typically perform a series of convolutions with these component impulse responses (Savioja et al., 1999; Zotkin et al., 2004; Sunder et al., 2015; Zhang et al., 2017). For a more detailed discussion, see Appendix A.4.

Given their success in speech synthesis (Wang et al., 2017), neural networks gained increased attention for audio generation recently. While most approaches focus on models in frequency domain (Choi et al., 2018; Vasquez & Lewis, 2019), raw waveform models were long neglected due to the difficulty to model long-range dependencies on a high-frequency audio signal. With the success of WaveNet (Van Den Oord et al., 2016) however, direct wave-to-wave modeling is of increasing interest (Fu et al., 2017; Luo & Mesgarani, 2018; Donahue et al., 2019) and shows major improvements in speech enhancement (Defossez et al., 2020) and denoising (Rethage et al., 2018), speech synthesis (Kalchbrenner et al., 2018), and music style translation (Mor et al., 2019).

More recently, first steps towards neural sound spatialization have been undertaken. Gebru et al. (2021) showed that HRTFs can be implicitly learned by neural networks trained on raw waveforms. Focusing on predicting spatial sound conditioned on visual information, a work by Morgado et al. (2018) aims to spatialize sound conditioned on $360°$ video. Yet, their work is limited to first order ambisonics and can not model detailed binaural effects. More closely related is a line of papers originating from the 2.5D visual sound system by Gao & Grauman (2019b). In this work, binaural audio is generated conditioned on a video frame embedding such that object locations can contribute to where sound comes from. Yang et al. (2020); Lu et al. (2019); Zhou et al. (2020) build upon the same idea. Unfortunately, all these works have in common that they pose the spatialization task as an upmixing problem, i.e., their models are trained with a mixture of left and right ear binaural recording as pseudo mono input. By design, these methods fail to model time delays and reverberation effects caused by the difference between source and listener position.

## 2  A NEURAL NETWORK FOR BINAURAL SYNTHESIS

We consider the problem where a monaural (single-channel) signal $x_{1:T} = (x_1, \ldots, x_T)$ of length $T$ is to be transformed into a binaural (stereophonic) signal $y_{1:T}^{(l)}, y_{1:T}^{(r)}$ representing the listener's left ear and right ear, given a conditioning temporal signal $\boldsymbol{c}_{1:T}$. This conditioning signal is the position and orientation of source and listener, respectively. Here $x_t$, and correspondingly $y_t^{(l)}$ and $y_t^{(r)}$, are scalars representing an audio sample at time $t$. In other words, we aim to produce a function,

$$\left(y_t^{(l)}, y_t^{(r)}\right) = f(x_{t-\Delta:t} | \boldsymbol{c}_{t-\Delta:t}),$$

where $\Delta$ is a temporal receptive field. Each $\boldsymbol{c}_t \in \mathbb{R}^{14}$ contains the 3D position of source and listener (three values each) and their orientations as quaternions (four values each). Note that in practice, $\boldsymbol{c}$

often is of lower frequency than the input and output signals $x_{1:T}$ and $y_{1:T}^{(l/r)}$ – source and listener positions would likely not be updated at 48kHz but rather at typical camera frame rates such as 30-120Hz. To simplify notation, we assume that $c$ has already been upsampled to the same temporal resolution as the audio signals.

Our overall framework is shown in Figure 1. A neural time warping module first warps the single-channel input signal $x_{1:T}$ into a two-channel signal $x_{1:T}^{(l/r)}$, where the channels represent left and right ear. The time warping compensates for coarse temporal effects and differences in time of sound arrival at left and right ear caused by the distance between source and listener. The second block in Figure 1 is a stack of $N$ layers, each of which is a conditioned hyper-convolution (see Section 2.2) followed by a sine activation, which has been shown to be beneficial for modeling higher frequencies (Sitzmann et al., 2020). Following the design of WaveNet, we use kernel size 2 and double the dilation factor in each layer to increase the receptive field. This temporal ConvNet models nuanced effects caused by room reverberations, head and ear shape, or changing head orientations.

## 2.1 NEURAL TIME WARPING

Time warping is the task of mapping a source temporal sequence onto a target sequence and has a long tradition in temporal signal processing. Most prominently, dynamic time warping (DTW) finds application in tasks like speech recognition (Juang, 1984) or audio retrieval (Deng & Leung, 2015). DTW can be characterized as finding a warpfield $\rho_{1:T}$ that warps a source signal $x_{1:T}$ to a target signal $\hat{x}_{1:T}$ such that the distance between the signals is minimized,

$$\rho_{1:T} = \underset{\tilde{\rho}_{1:T}}{\arg\min} \sum_t \|\hat{x}_t - x_{\tilde{\rho}_t}\|, \quad \text{where} \ \ \rho_t \in \{1, \ldots, T\}, \tag{1}$$

where the warpfield is typically constrained to respect physical properties such as monotonicity ($\rho_t \geq \rho_{t-1}$) and causality ($\rho_t \leq t$).

For binaural audio, there is a clear monotonous and causal relationship between source and target signal but the target signal is unknown at inference time. Additionally, the warping from mono to binaural signals goes far beyond simple linear time-shifts. For example, consider the source moving from the front to the left of the listener. This causes the delay between source and left ear to decrease but the delay between source and right ear to increase. If source and/or listener are moving, other wave effects such as the Doppler effect influence how the signal needs to be warped from the source to the listener's left and right ear. We are therefore interested in estimating a warpfield from the conditioning input $c_{1:T}$, i.e., from the spatial position and orientation of source and listener. A simple, parameter-free approach is geometric warping based on the speed of sound $\nu_{\text{sound}}$ and the distance between source and listener. Let $p_t^{(\text{src})}$ and $p_t^{(\text{lstn})}$ be the source and listener positions at time $t$ (which are part of $c_t$). Then,

$$\rho_t^{(\text{geom})} = t - \|p_t^{(\text{src})} - p_t^{(\text{lstn})}\| \cdot \frac{\text{audio sample rate}}{\nu_{\text{sound}}}. \tag{2}$$

This approach, however, fails to model important nuances such as the displacement between the left and right ear or diffraction delays as sound travels around the listener's head rather than straight through. In order to correct for those effects that geometric warping can not model properly, we estimate a neural warpfield $\rho_{1:T}^{(\text{neural})} = \text{WarpNet}(c_{1:T})$ and add it to the geometric warpfield (cf. Figure 1),

$$\rho_t = \sigma^{(\text{warp})}(\rho_{t-1}, \hat{\rho}_t) \quad \text{with} \quad \hat{\rho}_t := \rho_t^{(\text{neural})} + \rho_t^{(\text{geom})}, \tag{3}$$

where $\sigma^{(\text{warp})}(\rho_{t-1}, \hat{\rho}_t) = \max(\rho_{t-1}, \min(t, \hat{\rho}_t))$ is a recursive activation function that ensures monotonicity and causality. The WarpNet is a shallow temporal convolutional network with four layers and 64 channels each.

The warped signal can now be computed using the predicted warpfield. Since the warpfield elements $\rho_t$ are typically not integers, we define the warped signal $\hat{x}_{1:T}$ to be the linear interpolation of the original signal $x_{1:T}$ at positions $\lfloor \rho_t \rfloor$ and $\lceil \rho_t \rceil$,

$$\hat{x}_t = (\lceil \rho_t \rceil - \rho_t) \cdot x_{\lfloor \rho_t \rfloor} + (\rho_t - \lfloor \rho_t \rfloor) \cdot x_{\lceil \rho_t \rceil}. \tag{4}$$

In practice two warpfields are generated, one for each ear. Note how we explicitly enforce physical constraints in the warping by $\sigma^{(\text{warp})}$: $\min(t, \hat{\rho}_t)$ ensures causality by enforcing that the $t$-th element

of the warpfield can not be larger than $t$ itself. Monotonicity is enforced by $\max(\rho_{t-1}, \cdot)$: if an element has been warped from $\rho_{t-1}$ to position $t - 1$, the next element at position $t$ must be warped from $\rho_{t-1}$ or a succeeding position. In contrast to related approaches such as deformable convolutions (Dai et al., 2017) and spatial transformer networks (Jaderberg et al., 2015), our neural time warping therefore allows for constrained warping of input signals with arbitrary lengths and directly models a physical phenomenon of sound.

## 2.2 Conditioned hyper-convolutions

Raw waveform models where the output depends on an input signal and an additional conditioning temporal signal have primarily been studied in speech synthesis (Van Den Oord et al., 2016). The predominant approach towards such conditional temporal convolutions is an additive combination of the input signal $x_{1:T}$ and the conditioning signal $c_{1:T}$, i.e., $z_{1:T} = \mathbf{W} * x_{1:T} + \mathbf{V} * c_{1:T} + b$, such that the result of the convolution at time $t$ is

$$z_t = \sum_{k=1}^{K} \mathbf{W}_{:,:,k} x_{t-k+1} + \sum_{k=1}^{K} \mathbf{V}_{:,:,k} c_{t-k+1} + b. \tag{5}$$

Here, $\mathbf{W} \in \mathbb{R}^{C_{\text{out}} \times C_{\text{in}} \times K}$ and $\mathbf{V} \in \mathbb{R}^{C_{\text{out}} \times C_{\text{cond}} \times K}$ are tensors containing the weights for temporal convolutions of the $C_{\text{in}}$-dimensional input signal $x_{1:T}$ and the $C_{\text{cond}}$-dimensional conditional signal $c_{1:T}$ with a kernel size of $K$. Note that the convolutional weights $\mathbf{W}$ and $\mathbf{V}$ in this formulation are constant over time. Binaural filters in traditional digital signal processing, on the contrary, depend on the position of the sound source.
Inspired by the DSP formulation, we predict the convolutional weights for the input $x_{1:T}$ of a layer and the bias as functions of the conditioning input $c_{1:T}$,

$$z_t = \sum_{k=1}^{K} \left[ \mathcal{H}^{(\mathbf{W})}(c_{1:t}) \right]_{:,:,k} x_{t-k+1} + \mathcal{H}^{(b)}(c_{1:t}). \tag{6}$$

This formulation is similar to the use of hyper-networks in Ha et al. (2017) but rather than generating them from intermediate feature maps, weights are generated from the conditioning input $c_{1:T}$ that contains physical information about the relation between source and listener. $\mathcal{H}^{(\mathbf{W})}$ and $\mathcal{H}^{(b)}$ are small convolutional hyper-networks that receive $c_{1:t}$ as input and predict the convolutional weights and the bias as their output, respectively. Therefore, not only is the input to the convolutional layer a temporal sequence but the weights and biases change over time as well. We show in Appendix A.3 that if $\mathcal{H}^{(\mathbf{W})}$ and $\mathcal{H}^{(b)}$ are linear networks, hyper-convolutions equal equation 5 plus a biliear term.

## 2.3 Deficiencies of the $\ell_2$-loss on Raw Waveforms

Training a generative audio model with an $\ell_2$-loss on the raw waveform is generally considered to result in poor audio quality and distorted signals particularly for speech. Therefore, a number of mostly spectrogram oriented alternative loss functions have been introduced over recent years (Kolbæk et al., 2020). Here, we provide an analytical explanation for a fundamental problem of phase estimation with the $\ell_2$-loss on the waveform and show that a simple additional loss term mitigates the problem. While correct phase estimation is not critical for single-channel audio, it is crucial for binaural audio as our ears are sensitive to interaural time differences as small as $10\mu s$ (Brown & Duda, 1998). To start the analysis, let

$$\mathcal{L}_2(y_{1:T}, \hat{y}_{1:T}) = \sum_t (y_t - \hat{y}_t)^2 \tag{7}$$

be the time-domain $\ell_2$-loss between the predicted audio signal $y_{1:T}$ and the target $\hat{y}_{1:T}$ and let $Y_k, \hat{Y}_k \in \mathbb{C}$ denote the $k$-th frequency component of $y_{1:t}$ and $\hat{y}_{1:T}$ in the Fourier domain. We denote the amplitude error and angular phase error of the $k$-th frequency component as

$$\mathcal{L}^{(\text{amp})}(Y_k, \hat{Y}_k) = \left| |Y_k| - |\hat{Y}_k| \right| \quad \text{and} \quad \mathcal{L}^{(\text{phase})}(Y_k, \hat{Y}_k) = \angle(Y_k, \hat{Y}_k), \tag{8}$$

where $|\cdot|$ is the modulus (or magnitude) of the complex number.

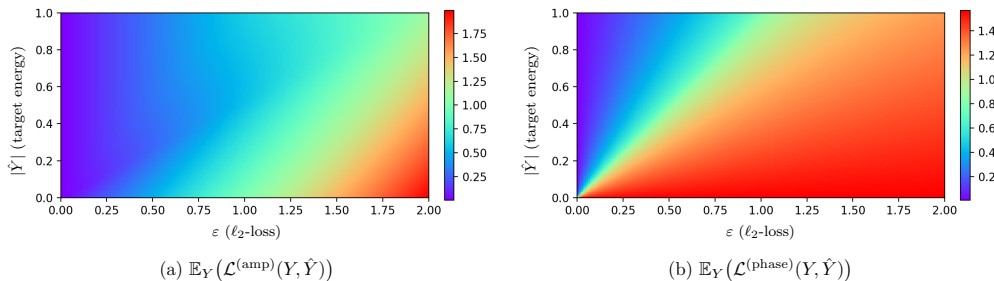

(a) $\mathbb{E}_Y\big(\mathcal{L}^{(\mathrm{amp})}(Y,\hat{Y})\big)$          (b) $\mathbb{E}_Y\big(\mathcal{L}^{(\mathrm{phase})}(Y,\hat{Y})\big)$

Figure 2: Expected amplitude and phase error from Lemma 1 as a function of $\ell_2$-value $\varepsilon$ and target signal energy $|\hat{Y}|$.

**Lemma 1.** *Let $\hat{Y} \in \mathbb{C}$ be a fixed complex number and $Y \in \mathbb{B}_{\varepsilon,\hat{Y}} = \{Y \in \mathbb{C} : |Y - \hat{Y}| = \varepsilon\}$ be any complex number that has distance $\varepsilon$ from $\hat{Y}$. Then, the expected amplitude error and the expected angular phase error with respect to $\hat{Y}$ are*

$$\mathbb{E}_Y\big(\mathcal{L}^{(\mathrm{amp})}(Y,\hat{Y})\big) = \frac{1}{2\pi}|\hat{Y}|\int_{-\pi}^{\pi}\left|\left|\frac{\varepsilon}{|\hat{Y}|}+e^{i\varphi}\right|-1\right|d\varphi \qquad \text{and} \tag{9}$$

$$\mathbb{E}_Y\big(\mathcal{L}^{(\mathrm{phase})}(Y,\hat{Y})\big) = \frac{1}{2\pi}\int_{-\pi}^{\pi}\arccos\frac{\mathrm{Re}\left(\frac{\varepsilon}{|\hat{Y}|}e^{i\varphi}+1\right)}{\left|\frac{\varepsilon}{|\hat{Y}|}+e^{i\varphi}\right|}d\varphi. \tag{10}$$

*Proof.* See Appendix A.1. $\qquad\qquad\qquad\qquad\qquad\qquad\qquad\qquad\qquad\qquad\qquad\qquad\qquad\qquad\square$

Using Parseval's theorem, we write the time-domain $\ell_2$-loss as the $\ell_2$-loss on the complex spectrum,

$$\mathcal{L}_2(y_{1:T}, \hat{y}_{1:T}) = \sum_k |Y_k - \hat{Y}_k|^2. \tag{11}$$

Now, consider a single summand from equation 11 and denote the distance $|Y_k - \hat{Y}_k|$ as $\varepsilon$. Lemma 1 allows us to analyze the expected amplitude and phase errors along this $k$-th frequency component. In Figure 2 we plot equation 9 and equation 10 as a function of the $\ell_2$-value $\varepsilon$ and the target energy $|\hat{Y}|$. There are two key insights. *First*, the expected amplitude error is low even for large $\ell_2$-values – that is, in the early stage of training – as long as the target signal has high energy (top right part of Figure 2a). The phase, on the contrary, is barely optimized at all early in training when the $\ell_2$-loss is large, even for high energy components, see Figure 2b. *Second*, over the course of training, i.e., when the $\ell_2$-loss decreases over time, the expected amplitude error among all target energies decreases. The expected phase error, on the other hand, improves primarily for high energy components and mid- and low energy components tend to have poor phase accuracy even for small $\ell_2$-values.

The above analysis shows that optimizing raw waveforms with a time-domain $\ell_2$-loss leads to a strong focus on fitting the amplitudes but accurate phase reconstruction falls short. Since the models have limited capacity, the training data usually can only be fit up to an $\ell_2$-loss $\varepsilon_{\min}$. If this $\varepsilon_{\min}$ is not sufficiently small, the signal's amplitude can be modeled well but phase errors will always be significant. This can be critical since small amplitude errors lead to a slight change in speech coloration but phase errors introduce perceivable distortions. To overcome the deficiencies of the time-domain $\ell_2$-loss in phase optimization, we add an explicit phase term to the loss function,

$$\mathcal{L}(y_{1:T}, \hat{y}_{1:T}) = \mathcal{L}_2(y_{1:T}, \hat{y}_{1:T}) + \lambda\mathcal{L}^{(\mathrm{phase})}\big(\mathrm{STFT}(y_{1:T}), \mathrm{STFT}(\hat{y}_{1:T})\big), \tag{12}$$

where $\mathrm{STFT}(y_{1:T})$ is the short-term Fourier transform of the audio signal $y_{1:T}$.[2]

---

[2]We discuss an alternative formulation of this loss that operates fully in frequency domain in Appendix A.2.

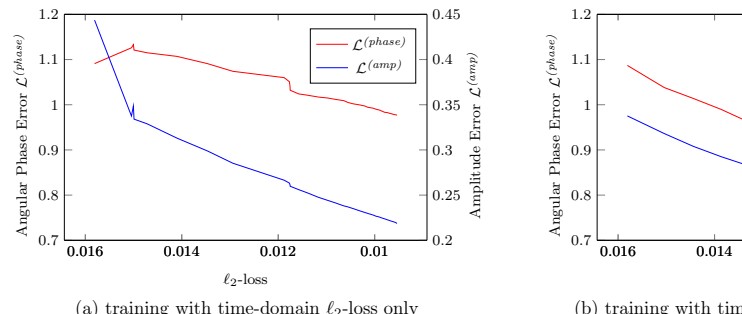

(a) training with time-domain $\ell_2$-loss only

(b) training with time-domain $\ell_2$-loss + phase loss

Figure 3: Development of phase- and amplitude error as the $\ell_2$-loss decreases during training.

Table 1: Comparison of commonly used losses for audio modeling to our proposed $\ell_2$ + phase loss.

|  | raw waveform ($\ell_2$ error $\times 10^3$) | power spectrum ($\ell_2$ error) | phase spectrum (angular error) |
|---|---|---|---|
| power spectrum + phase copy | 1.276 | 0.048 | 1.563 |
| multiscale STFT | 2.279 | 0.043 | 1.996 |
| Si-SDR | 0.798 | 0.222 | 1.507 |
| cross entropy on $\mu$-law encoding | 0.161 | 0.039 | 1.199 |
| $\ell_2$ | **0.141** | **0.037** | 0.886 |
| $\ell_2$ + phase loss (equation 12) | 0.167 | 0.048 | **0.807** |

## 3 EVALUATION

**Dataset.** We recorded a total of 2 hours of paired mono and binaural data at 48kHz from eight different speakers, four male and four female. The listener is a mannequin equipped with binaural microphones in its ears. Participants were asked to walk around the mannequin an a circle with 1.5m radius and have an unscripted conversation with it. We used an OptiTrack system to track position and orientation of source and listener throughout the captures. To the best of our knowledge, this is the only in-the-wild (i.e., not recorded in an anechoic chamber) binaural dataset of such size. We use a validation sequence and the last two minutes from each participant as test data and train the models on the remaining data. See Appendix A.5 for a more detailed description.

**Network Architecture.** The WarpNet architecture is as described in Section 2.1. The temporal convolutional network consists of three sequential blocks. Each block is a stack of ten hyper-convolution layers with 64 channels, kernel size 2, and the dilation size is doubled after each layer. We train our models for 100 epochs using an Adam optimizer. Learning rates are decreased if between two epochs the loss on the training set did not improve. At inference, our model can produce binaural audio in real-time.

### 3.1 LOSS EVALUATION

In order to empirically validate our findings from Section 2.3, we train our proposed network with time-domain $\ell_2$-loss only and with the loss proposed in equation 12. Figure 3 shows how the phase error and amplitude error develop during training as the time-domain $\ell_2$-loss decreases. The model trained with $\ell_2$-loss only (Figure 3a) shows the behaviour that the analysis in Section 2.3 suggests: the amplitude is optimized aggressively, particularly in the beginning in training when the $\ell_2$-loss is still high. The phase, on the contrary, does hardly improve at all in the beginning and shows only moderate improvements as the $\ell_2$-loss becomes smaller. When training with time-domain $\ell_2$-loss and phase loss (Figure 3b), this effect is being compensated for. The amplitude is optimized less aggressively and phase improves from the beginning of training on.

Various audio losses have been proposed over time, ranging from optimizing the power spectrum only and copying the input's phase (Zhao et al., 2018; Gao & Grauman, 2019a) over a multiscale STFT loss for high frequency and high time resolution (Yamamoto et al., 2020) to optimization of the scale-invariant signal to distortion ratio (si-SDR, Le Roux et al. (2019); Heitkaemper et al. (2020); Luo & Mesgarani (2019)). With the introduction of WaveNet for speech synthesis (Van Den Oord

Table 2: Ablation study. The components of the proposed binauralization network improve phase and amplitude and thereby the overall loss in time-domain.

|   |   | raw waveform ($\ell_2$ error $\times 10^3$) | power spectrum ($\ell_2$ error) | phase spectrum (angular error) |
|---|---|---|---|---|
| (a) | vanilla temporal CNN | 0.254 | 0.061 | 0.934 |
| (b) | + warping | 0.206 | 0.061 | 0.849 |
| (c) | + hyper-conv | 0.183 | 0.051 | 0.847 |
| (d) | + sine activation | **0.167** | **0.048** | **0.807** |

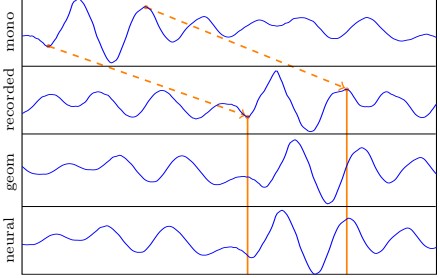

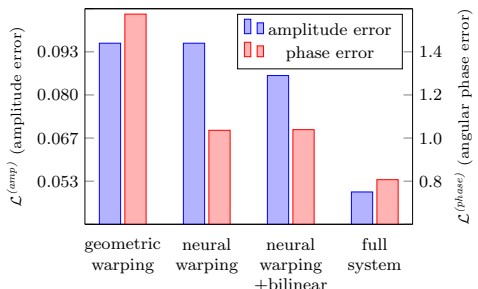

(a) Warping example. Top to bottom: source mono input; left ear binaural recording; geometric warping as in equation 2; neural time warping as in equation 4.

(b) Amplitude and phase error for different warping schemes, warping plus bilinear amplitude scale, and the full system.

Figure 4: Analysis of the warping module.

et al., 2016), interpreting audio optimization as categorical optimization on a $\mu$-law encoded signal has become a prominent technique. As Table 1 shows, all these approaches fail to predict accurate phase and mostly result in meager power spectral and waveform optimization. Overall, our proposed loss retains accurate $\ell_2$ and power spectral estimations while outperforming other criteria by a huge margin in phase error.

Perceptually, we observe a strong correlation between the phase error and noise and distortions in the generated binaural signal. In particular, our proposed loss was the only one that produced clean speech without perceivable distortions. This is consistent with our perceptual study in Table 4, where other approaches with different losses and architectures have been ranked less favorable.

## 3.2 MODEL EVALUATION

**Ablation Study.** In Table 2, we show the impact of our model's individual components compared to a vanilla temporal convolutional network baseline with a WaveNet-like architecture and ReLU activations. Number of layers, channels, and kernel sizes are the same as in our final system. Keeping amplitudes unchanged but compensating for interaural time differences, it is not surprising that neural time warping leads to a huge improvement in phase. Replacing regular convolutions with hyper-convolutions, on the contrary, is particularly beneficial to improve the power spectrum. Finally, replacing the ReLU activations by sine functions, which have been proven to retain high frequency details more reliably (Sitzmann et al., 2020), leads to an additional moderate improvement along waveform, phase, and amplitude error.

**Neural Time Warping.** The purpose of neural time warping is a strong initial alignment of the mono source signal to the left and right ear listener signal, respectively. Note the significant temporal shift between the mono signal and recorded left ear signal in Figure 4a. In the same figure, observe how geometric warping provides an approximate alignment to the reference signal, while the learned neural warping successfully corrects the inaccurate geometric warping and aligns the peaks and valleys more accurately. Although those adjustments seem small, the impact of neural warping on the phase error is significant, as shown in Figure 4b (red bars). Naturally, neural warping can not improve the amplitude (blue bars).

**Temporal HyperConv Network.** Neural warping provides an accurate alignment between input and target signal. This raises the question if a deep network is required on top of the warping

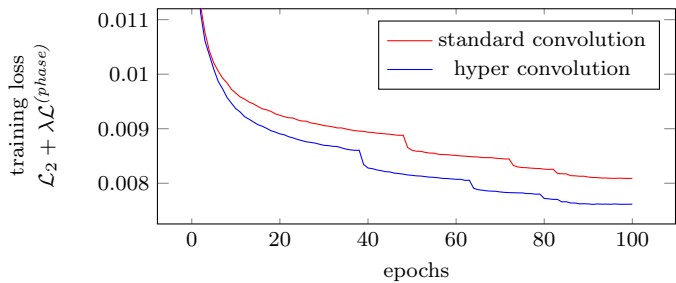

Figure 5: Training loss of a model with hyper-convolutions and a model with standard convolutions. Hyper-convolutions lead to a significantly faster convergence.

Table 3: Comparison to state of the art approaches for binaural sound synthesis.

| | raw waveform ($\ell_2$ error $\times 10^3$) | power spectrum ($\ell_2$ error) | phase spectrum (angular error) |
|---|---|---|---|
| DSP | 0.485 | 0.058 | 1.388 |
| 2.5D Sound | 1.085 | 0.113 | 1.519 |
| WaveNet | 0.237 | 0.048 | 1.239 |
| ours | **0.167** | **0.048** | **0.807** |

Table 4: Mean opinion scores of different approaches. Participants were ask to rank cleanliness, spatialization, and overall realism on a Likert scale from 1 to 5.

| | cleanliness | spatialization | realism |
|---|---|---|---|
| DSP | 3.48 $\pm$ 0.88 | 3.75 $\pm$ 0.98 | 3.62 $\pm$ 0.90 |
| 2.5D Sound | 2.70 $\pm$ 1.09 | 3.18 $\pm$ 0.94 | 2.70 $\pm$ 1.03 |
| WaveNet | 1.20 $\pm$ 0.51 | 2.92 $\pm$ 1.11 | 1.39 $\pm$ 0.71 |
| ours | **4.26** $\pm$ 0.89 | **3.76** $\pm$ 0.91 | **3.88** $\pm$ 0.99 |
| binaural recordings | 3.69 $\pm$ 0.94 | 3.88 $\pm$ 0.96 | 3.82 $\pm$ 0.88 |

module or if a linear amplitude adjustment can already yield convincing results. We therefore apply a learned bilinear term to the warped result,

$$y_t^{(l/r)} = x_t^{\text{(warped)}} \boldsymbol{a}^T \boldsymbol{c}_t + b, \qquad \boldsymbol{a} \in \mathbb{R}^{C_{\text{cond}}}, b \in \mathbb{R} \tag{13}$$

given the conditioning $\boldsymbol{c}_t$ and the warped signal $x^{\text{(warped)}}$ for the left or right ear, respectively. Figure 4b shows that this leads to a slight improvement of the amplitude error but falls way behind the performance of the full system with a deep temporal network of hyper-convolutions after the warping module. Inspection of the mono and recorded signal in Figure 4a in fact reveals that the binaural recording undergoes additional transformations beyond warping. Room reverberations, source speech directivity, and modifications caused by the shape of the listener's ear, for instance, are physical effects that require complex transformations of the warped signal.

Many of these subtle effects depend on the position and orientation of source and listener in the room. It is therefore plausible that conditioned hyper-convolutions, which can model more complex dependencies between inputs and conditioning variables in a single layer, show better performance than standard convolutions, cf. Table 2 (b) versus (c). As Figure 5 reveals, hyper-convolutions also converge significantly faster than standard convolutions in the early stages of training.

## 3.3 STATE OF THE ART COMPARISON

In Table 3, we compare our approach to other neural binauralization approaches and to a DSP baseline, which is the de-facto state of the art for binauralization. The recently proposed 2.5D visual sound (Gao & Grauman, 2019b) network operates in frequency domain and predicts a complex mask which the input is multiplied with to obtain left and right ear outputs. We compare to their approach and replace the visual features with our conditioning features $\boldsymbol{c}_{1:T}$. For the STFT, we use a window size of 1,600 samples and a hop length of 480 samples (10ms). Therefore, modeling delays of less than 10ms requires non-trivial manipulation of the phase information in the complex

Table 5: Real-time-factor for offline processing and latency for streaming generation of binaural audio. The DSP baseline runs on CPU, all other models run on an NVidia Tesla V100.

| | trainable parameters | offline inference *real-time-factor* | streaming mode *latency* | |
|---|---|---|---|---|
| DSP (* on CPU) | – | 0.680 | 25.0ms | ($\pm$2.1ms) |
| 2.5D Sound | 16.7M | 0.013 | – | |
| WaveNet | 1.9M | 0.043 | 31.9ms | ($\pm$0.3ms) |
| ours | 8.6M | 0.069 | 32.8ms | ($\pm$0.4ms) |

spectrogram, which is a more difficult operation than modeling delays in time-domain. We also provide a comparison to a WaveNet that proved to be generally strong in various generative audio problems (Rethage et al., 2018; Engel et al., 2017). For the conditioning on source and listener positions, we follow the approach of Van Den Oord et al. (2016), i.e., the source and listener positions are appended to the input of each temporal convolutional layer. Note that we use the WaveNet in a non-autoregressive setup since the input audio, i.e., the mono signal, is fully available at inference time. Overall, our approach performs significantly better than other methods.

For a perceptual evaluation, we asked 100 participants to rank a total of 2,000 audio snippets from 1 to 5 on a Likert scale according to three criteria: cleanliness of the signal, spatialization quality, and overall realism, see Table 4. All scores are below a 5 (indistinguishable from reality) because participants listened to results for a generic head-related transfer function rather to one that takes their explicit head and ear shape into account. Additionally, user's headphones are of different quality and not equalized. Note that the binaural (ground truth) recordings score lower on cleanliness because they contain ambient noise that is uncorrelated to the source input and therefore not modeled by our approach. WaveNet leads to a particularly noisy audio signal, which is caused by two major factors. First, audio is generated at 48kHz, which is more difficult to model than 16-24kHz audio. Arik et al. (2017) show that the quality of WaveNet degrades with higher sampling rates. Second, WaveNet has to spend a considerable amount of capacity on modeling large source-to-listener time shifts between the mono and binaural signals and, in consequence, struggles more to generate clean audio. Our approach ranks favorably against other neural binauralization systems and is also preferred in terms of cleanliness and realism over the DSP baseline. A t-test showed that all results in Table 4 are statistically significant with the exception of spatialization between ours and DSP, which is ranked at almost equal quality. Since DSP is the perceptually closest competitor to our approach, we performed an additional perceptual side-by-side study between the two systems that comfirms the results presented in Table 4, see Appendix A.6.

**Runtime.** When analyzing the runtime of a system, two cases are important. The first is *offline processing*, where a user provides the complete mono audio to be binauralized in advance. The real-time-factor is the computation time divided by the duration of the input. Table 5 shows that our system allows for a rapid binauralization. On a single NVidia Tesla V100, our approach can binauralize 100 seconds of mono audio in just 6.9 seconds. Note that the DSP baseline does not run on a GPU but is purely CPU-based. The second case is a *streaming scenario* where binaural audio has to be computed on-the-fly, e.g., when a user navigates through a 3D environment in a game or in virtual reality. In this case, systems are required to have low latency. On an NVidia Tesla V100, our approach runs with roughly 33ms latency, which is low enough to have non-observable delays for videos or games that render frames at 30Hz. Note that this is measured with a naive pytorch implementation and allows for several improvements to further lower the latency. The 2.5D sound network can not efficiently be applied in a streaming mode because (a) it is acausal, i.e., requires access to future audio, and (b) it summarizes as much as 0.32 seconds of audio in a single temporal step in its bottleneck layer due to the UNet structure.

## 4 CONCLUSION

Our neural sound binauralization approach is the first purely data-driven end-to-end model that shows convincing performance compared to traditional state of the art binauralization methods. We were able to show effectiveness of our model both quantitatively and in a perceptual user study. Moreover, we unveiled and mitigated a fundamental issue with $\ell_2$-optimization on the raw waveform that affects not only this task but is relevant to other generative audio problems as well.

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

## A    APPENDIX

### A.1    PROOF OF LEMMA 1

Before we start with the formal proof, let us get a more intuitive idea what Lemma 1 means and how phase and amplitude error relate to the $\ell_2$-loss in time-domain and complex spectral domain, respectively. Reconsider the $\ell_2$-loss in time-domain and use Parseval's theorem to relate it to the complex frequency domain,

$$\sum_t (y_t - \hat{y}_t)^2 = \sum_k |Y_k - \hat{Y}_k|^2, \tag{14}$$

where $k$ runs over all frequency components of the signal. The time-domain $\ell_2$-loss is therefore a sum of the $\ell_2$-loss of each individual frequency component in spectral domain. For the analysis, let us consider one fixed frequency component $k$. Our findings hold for all frequency components equally. Figure 6 illustrates this case in the complex plane.[3] Given a target $\hat{Y}$ and a prediction $Y$ that has distance $\varepsilon$ to $\hat{Y}$, optimizing the $\ell_2$-loss is equal to optimization of the amplitude (the length difference of $Y$ and $\hat{Y}$) if $\theta_Y = \theta_{\hat{Y}}$. If the phases $\theta_Y$ and $\theta_{\hat{Y}}$ are different, however, the relation is not that obvious. Lemma 1 makes a statement about the expected amplitude and phase error for a random prediction $Y$ that has $\ell_2$-distance $\varepsilon$ to $\hat{Y}$, i.e., that lies on the circle defined by $\mathbb{B}_{\varepsilon,\hat{Y}}$.

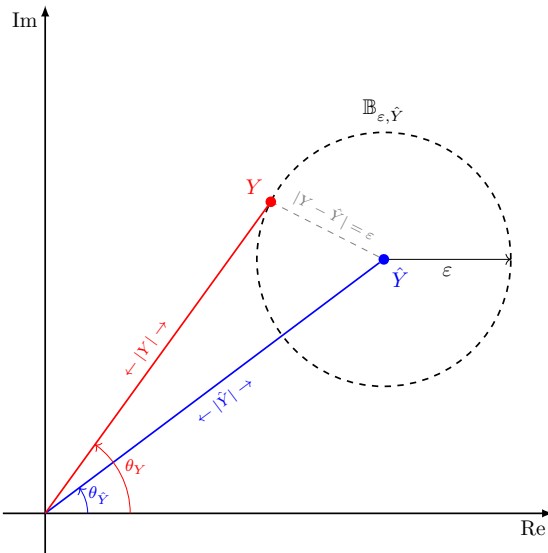

Figure 6: Graphical illustration of the premises for Lemma 1 on the complex plane. $\hat{Y}$ is the target, $Y$ is a prediction with distance $\varepsilon$ to $\hat{Y}$. The amplitude error is defined as $\big||Y| - |\hat{Y}|\big|$ and the phase error is the difference between $\theta_Y$ and $\theta_{\hat{Y}}$.

With this in mind, we prove

**Lemma 1.** *Let $\hat{Y} \in \mathbb{C}$ be a fixed complex number and $Y \in \mathbb{B}_{\varepsilon,\hat{Y}} = \{Y \in \mathbb{C} : |Y - \hat{Y}| = \varepsilon\}$ be any complex number that has distance $\varepsilon$ from $\hat{Y}$. Then, the expected amplitude error and the expected*

---

[3]We drop the index $k$ and just write $Y$ and $\hat{Y}$.

*angular phase error with respect to $\hat{Y}$ are*

$$
\begin{aligned}
\mathbb{E}_Y\big(\mathcal{L}^{(\mathrm{amp})}(Y,\hat{Y})\big) &= \int_{Y\in\mathbb{B}_{\varepsilon,\hat{Y}}} \mathcal{L}^{(\mathrm{amp})}(Y,\hat{Y})p(Y)dY \\
&= \frac{1}{2\pi}|\hat{Y}|\int_{-\pi}^{\pi}\Big|\,\big|\frac{\varepsilon}{|\hat{Y}|}+e^{i\varphi}\big|-1\Big|d\varphi \qquad \textit{and}
\end{aligned}
\tag{15}
$$

$$
\begin{aligned}
\mathbb{E}_Y\big(\mathcal{L}^{(\mathrm{phase})}(Y,\hat{Y})\big) &= \int_{Y\in\mathbb{B}_{\varepsilon,\hat{Y}}} \mathcal{L}^{(\mathrm{phase})}(Y,\hat{Y})p(Y)dY \\
&= \frac{1}{2\pi}\int_{-\pi}^{\pi}\arccos\frac{\mathrm{Re}\Big(\frac{\varepsilon}{|\hat{Y}|}e^{i\varphi}+1\Big)}{\big|\frac{\varepsilon}{|\hat{Y}|}+e^{i\varphi}\big|}d\varphi.
\end{aligned}
\tag{16}
$$

*Proof.* Let $\hat{Y}\in\mathbb{C}$ and $Y\in\mathbb{B}_{\varepsilon,\hat{Y}}$. Then, amplitude loss and phase loss as defined in equation 8 are given by

$$
\mathcal{L}^{(\mathrm{amp})}(Y,\hat{Y}) = \big|\,|Y|-|\hat{Y}|\,\big|,
\tag{17}
$$

$$
\mathcal{L}^{(\mathrm{phase})}(Y,\hat{Y}) = \angle(Y,\hat{Y}) = \arccos\frac{\mathrm{Re}(Y)\mathrm{Re}(\hat{Y})+\mathrm{Im}(Y)\mathrm{Im}(\hat{Y})}{|Y|\cdot|\hat{Y}|},
\tag{18}
$$

where $|\cdot|$ denotes the modulus (or magnitude) of a complex number.

Without loss of generality, let $\mathrm{Im}(\hat{Y})=0$ and $\mathrm{Re}(\hat{Y})\geq 0$. This can always be achieved by a unitary rotation of angle $\theta_{\hat{Y}}$ around the origin of the complex plane which preserves the lengths and angles between $Y$ and $\hat{Y}$. In that case, the phase loss simplifies to

$$
\mathcal{L}^{(\mathrm{phase})}(Y,\hat{Y}) = \arccos\frac{\mathrm{Re}(Y)\cdot|\hat{Y}|}{|Y|\cdot|\hat{Y}|} = \arccos\frac{\mathrm{Re}(Y)}{|Y|}.
\tag{19}
$$

Since $Y\in\mathbb{B}_{\varepsilon,\hat{Y}}$ is a point on the circle around $\hat{Y}$ with radius $\varepsilon$ (see Figure 6 for an illustration), $\mathbb{B}_{\varepsilon,\hat{Y}}$ can equivalently be written as

$$
\mathbb{B}_{\varepsilon,\hat{Y}} = \{Y\in\mathbb{C}:Y=\varepsilon\cdot e^{i\varphi}+\hat{Y},\varphi\in[-\pi,\pi]\}.
\tag{20}
$$

Then, each $Y\in\mathbb{B}_{\varepsilon,\hat{Y}}$ is uniquely defined by some $\varphi\in[-\pi,\pi]$ and we can rewrite $\mathcal{L}^{(amp)}$ as a function of $\varphi$ by substituting $Y=\varepsilon e^{i\varphi}+\hat{Y}$,

$$
\mathcal{L}^{(\mathrm{amp})}_{\varepsilon,\hat{Y}}(\varphi) = \big|\,|\varepsilon e^{i\varphi}+\hat{Y}|-|\hat{Y}|\,\big|.
\tag{21}
$$

Observe that due to $\mathrm{Im}(\hat{Y})=0$ and $\mathrm{Re}(\hat{Y})\geq 0$, it follows that $\mathrm{Re}(\hat{Y})=|\hat{Y}|$ and

$$
\begin{aligned}
\big|\varepsilon e^{i\varphi}+\hat{Y}\big| &= \sqrt{(\varepsilon\cos\varphi+|\hat{Y}|)^2+\varepsilon^2\sin(\varphi)^2} \\
&= |\hat{Y}|\sqrt{\frac{\varepsilon^2}{|\hat{Y}|^2}+2\frac{\varepsilon}{|\hat{Y}|}\cos\varphi+1} \\
&= |\hat{Y}|\sqrt{\frac{\varepsilon^2}{|\hat{Y}|^2}+2\frac{\varepsilon}{|\hat{Y}|}\cos\varphi+\cos(\varphi)^2+\sin(\varphi)^2} \\
&= |\hat{Y}|\cdot\big|\frac{\varepsilon}{|\hat{Y}|}+e^{i\varphi}\big|.
\end{aligned}
\tag{22}
$$

Therefore,

$$
\mathcal{L}^{(\mathrm{amp})}_{\varepsilon,\hat{Y}}(\varphi) = |\hat{Y}|\cdot\Big|\,\big|\frac{\varepsilon}{|\hat{Y}|}+e^{i\varphi}\big|-1\Big|.
\tag{23}
$$

For the phase error, we apply the same substitution and obtain

$$
\begin{aligned}
\mathcal{L}^{(\text{phase})}_{\varepsilon,\hat{Y}}(\varphi) &= \arccos \frac{\text{Re}(\varepsilon e^{i\varphi} + \hat{Y})}{|\varepsilon e^{i\varphi} + \hat{Y}|} \\
&= \arccos \frac{|\hat{Y}|\big(\frac{\varepsilon}{|\hat{Y}|}\cos\varphi + 1\big)}{|\hat{Y}| \cdot \big|\frac{\varepsilon}{|\hat{Y}|} + e^{i\varphi}\big|} \\
&= \arccos \frac{\frac{\varepsilon}{|\hat{Y}|}\cos\varphi + 1}{\big|\frac{\varepsilon}{|\hat{Y}|} + e^{i\varphi}\big|} \\
&= \arccos \frac{\text{Re}\big(\frac{\varepsilon}{|\hat{Y}|}e^{i\varphi} + 1\big)}{\big|\frac{\varepsilon}{|\hat{Y}|} + e^{i\varphi}\big|}.
\end{aligned}
\tag{24}
$$

The expected amplitude and phase error can now be written as

$$
\mathbb{E}_\varphi(\mathcal{L}^{(\text{amp})}_{\varepsilon,\hat{Y}}(\varphi)) = \int_{-\pi}^{\pi} \mathcal{L}^{(\text{amp})}_{\varepsilon,\hat{Y}}(\varphi)p(\varphi)d\varphi \qquad \text{and}
\tag{25}
$$

$$
\mathbb{E}_\varphi(\mathcal{L}^{(\text{phase})}_{\varepsilon,\hat{Y}}(\varphi)) = \int_{-\pi}^{\pi} \mathcal{L}^{(\text{phase})}_{\varepsilon,\hat{Y}}(\varphi)p(\varphi)d\varphi.
\tag{26}
$$

Assuming that $Y \in \mathbb{B}_{\varepsilon,\hat{Y}}$ is uniformly distributed on the circle around $\hat{Y}$, i.e., $p(\varphi)$ follows a circular uniform distribution, we have $p(\varphi) = \frac{1}{2\pi}$ and plug in equation 23 and equation 24 to obtain the claim from Lemma 1. □

## A.2 Frequency-Domain Loss Formulation

In our original formulation of the phase-enhanced loss in equation 12, phase is technically penalized twice: once implicitly in the $\ell_2$-loss on the raw waveform and once explicitly in the phase loss term. It is possible to formulate the phase-enhanced loss in frequency domain such that there is a clear separation of magnitude and phase loss in two different additive terms.

Due to Parseval's theorem, optimizing the $\ell_2$-loss in time domain is equivalent to optimizing the $\ell_2$-loss in frequency domain. We transform the time domain audio signal into frequency domain using a short-term Fourier transform and denote the result of the transformation as

$$
Y_{1:K,1:S} = \text{STFT}(y_{1:T}),
\tag{27}
$$

where $K$ is the number of frequency bins of the discrete Fourier transform and $S$ is the number of STFT steps, i.e., $S = T/\text{hop\_length}$. Since the frequency spectrum can be expressed by it's magnitude and phase spectrum, we can reformulate equation 12 as

$$
\mathcal{L}(y_{1:T}, \hat{y}_{1:T}) = \sum_{k,s} \Big[ \mathcal{L}^{(\text{amp})}(Y_{k,s}, \hat{Y}_{k,s}) + \lambda \mathcal{L}^{(\text{phase})}(Y_{k,s}, \hat{Y}_{k,s}) \Big].
\tag{28}
$$

Note that this form is not equivalent to the original loss since separating the frequency spectrum into magnitude and phase is a non-unitary operation. However, we do not find the differences to be significant, as Table 6 shows. In fact, the clear separation allows for a better adjustment of the phase weight $\lambda$ and can even lead to slightly improved results.

## A.3 Interpretation of hyper-convolutions

Hyper-convolutions are a generalization of stardard convolutions as long as $\mathcal{H}^{(\mathbf{W})}$ and $\mathcal{H}^{(\boldsymbol{b})}$ are able to learn a linear transformation. Specifically, if $\mathcal{H}^{(\mathbf{W})}$ and $\mathcal{H}^{(\boldsymbol{b})}$ are linear convolutional networks, hyper-convolutions extend equation 5 by a bilinear term. We show this property in the following and provide an example for a simple fully connected layer.

Recall the definition of hyper-convolutions from equation 6,

$$
\boldsymbol{z}_t = \sum_{k=1}^{K} \big[\mathcal{H}^{(\mathbf{W})}(\boldsymbol{c}_{1:t})\big]_{:,:,k}\boldsymbol{x}_{t-k+1} + \mathcal{H}^{(\boldsymbol{b})}(\boldsymbol{c}_{1:t}).
\tag{29}
$$

Table 6: Comparison of the loss formulation from equation 12 and equation 28. While the first penalizes phase twice, once implicitly in the time-domain $\ell_2$-loss and once in the explicit phase loss term, the latter provides a clear separation between magnitude and phase loss terms.

|  | raw waveform ($\ell_2$ error $\times 10^3$) | power spectrum ($\ell_2$ error) | phase spectrum (angular error) |
|---|---|---|---|
| loss from equation 12 | 0.167 | 0.048 | 0.807 |
| loss from equation 28 | 0.157 | 0.036 | 0.809 |

**Lemma 2.** *Let $\mathcal{H}^{(\mathbf{W})}$ and $\mathcal{H}^{(\boldsymbol{b})}$ be linear convolutional networks with kernel size $K$. Let further $\boldsymbol{x}_{1:T}$ and $\boldsymbol{c}_{1:T}$ be a sequence of input and conditional vectors in $\mathbb{R}^{C_{in}}$ and $\mathbb{R}^{C_{cond}}$, and the output $\boldsymbol{z}_{1:T}$ be a sequence of vectors in $\mathbb{R}^{C_{out}}$. Then, the hyper-convolution from equation 29 reduces to*

$$\boldsymbol{z}_t = \sum_{k=1}^{K}\sum_{k'=1}^{K}\sum_{j=1}^{C_{in}} x_{t-k+1,j} \cdot \mathbf{U}_{:,j,k,:,k'}\boldsymbol{c}_{t-k'+1}$$
$$+ \sum_{k=1}^{K}\mathbf{W}_{:,:,k}\boldsymbol{x}_{t-k+1:t} + \sum_{k=1}^{K}\mathbf{V}_{:,:,k}\boldsymbol{c}_{t-k+1} + \boldsymbol{b} \tag{30}$$

*with $\mathbf{U} \in \mathbb{R}^{(C_{out}\times C_{in}\times K)\times(C_{cond}\times K)}$, $\mathbf{W} \in \mathbb{R}^{C_{out}\times C_{in}\times K}$, $\mathbf{V} \in \mathbb{R}^{C_{out}\times C_{cond}\times K}$ and $\boldsymbol{b} \in \mathbb{R}^{C_{out}}$.*

*Proof.* We start from equation 29 and use that both $\mathcal{H}^{(\mathbf{W})}$ and $\mathcal{H}^{(\boldsymbol{b})}$ are linear (and therefore, w.l.o.g. single-layer) convolutional networks with kernel size $K$, such that

$$\boldsymbol{z}_t = \sum_{k=1}^{K}\left[\mathcal{H}^{(\mathbf{W})}(\boldsymbol{c}_{t-K+1:t})\right]_{:,:,k}\boldsymbol{x}_{t-k+1} + \mathcal{H}^{(\boldsymbol{b})}(\boldsymbol{c}_{t-K+1:t}). \tag{31}$$

First, consider $\mathcal{H}^{(\boldsymbol{b})} : \mathbb{R}^{C_{cond}\times K} \mapsto \mathbb{R}^{C_{out}}$. Since $\mathcal{H}^{(\boldsymbol{b})}$ is linear, it can be written as

$$\mathcal{H}^{(\boldsymbol{b})}(\boldsymbol{c}_{t-K+1:t}) = \sum_{k=1}^{K}\mathbf{V}_{:,:,k}\boldsymbol{c}_{t-k+1} + \boldsymbol{b} \tag{32}$$

with $\mathbf{V} \in \mathbb{R}^{C_{out}\times C_{cond}\times K}$ and $\boldsymbol{b} \in \mathbb{R}^{C_{out}}$. Note that this already yields the last two terms in equation 30. While $\mathcal{H}^{(\boldsymbol{b})}$ only generates a bias, $\mathcal{H}^{(\mathbf{W})}$ needs to generate a $\mathbb{R}^{C_{out}\times C_{in}\times K}$ sized tensor to realize the convolution with $\boldsymbol{x}_{1:T}$. Thus, $\mathcal{H}^{(\mathbf{W})} : \mathbb{R}^{C_{cond}\times K} \mapsto \mathbb{R}^{C_{out}\times C_{in}\times K}$ and due to $\mathcal{H}^{(\mathbf{W})}$ being a linear function, the component at $(i,j,k)$ of the output tensor is given as

$$\left[\mathcal{H}^{(\mathbf{W})}(\boldsymbol{c}_{t-K+1:t})\right]_{i,j,k} = \sum_{k'=1}^{K}\mathbf{U}_{i,j,k,:,k'}\boldsymbol{c}_{t-k'+1} + \mathbf{W}_{i,j,k}, \tag{33}$$

where $\mathbf{U} \in \mathbb{R}^{(C_{out}\times C_{in}\times K)\times(C_{cond}\times K)}$ is the weight tensor of the hypernetwork $\mathcal{H}^{(\mathbf{W})}$ and $\mathbf{W} \in \mathbb{R}^{C_{out}\times C_{in}\times K}$ is its bias. For simplicity of notation, let $\hat{\boldsymbol{z}}_t$ be $\boldsymbol{z}_t$ from equation 31 without $\mathcal{H}^{(\boldsymbol{b})}(\boldsymbol{c}_{t-K+1:t})$. Using equation 33, $\hat{\boldsymbol{z}}_t$ is then given by

$$\hat{\boldsymbol{z}}_t = \sum_{k=1}^{K}\sum_{j=1}^{C_{in}}\left[\mathcal{H}^{(\mathbf{W})}(\boldsymbol{c}_{t-K+1:t})\right]_{:,j,k} \cdot x_{t-k+1,j}$$
$$= \sum_{k=1}^{K}\sum_{j=1}^{C_{in}}\left[\sum_{k'=1}^{K}\mathbf{U}_{:,j,k,:,k'}\boldsymbol{c}_{t-k'+1} + \mathbf{W}_{:,j,k}\right] \cdot x_{t-k+1,j}$$
$$= \sum_{k=1}^{K}\sum_{k'=1}^{K}\sum_{j=1}^{C_{in}} x_{t-k+1,j} \cdot \mathbf{U}_{:,j,k,:,k'}\boldsymbol{c}_{t-k'+1} + \sum_{k=1}^{K}\mathbf{W}_{:,:,k}\boldsymbol{x}_{t-k+1:t}. \tag{34}$$

Together with equation 32, this yields the claim from Lemma 2,

$$
\begin{aligned}
\boldsymbol{z}_t =& \hat{\boldsymbol{z}}_t + \mathcal{H}^{(\boldsymbol{b})}(\boldsymbol{c}_{t-K+1:t}) \\
=& \sum_{k=1}^{K}\sum_{k'=1}^{K}\sum_{j=1}^{C_{\text{in}}} x_{t-k+1,j} \cdot \mathbf{U}_{:,j,k,:,k'} \boldsymbol{c}_{t-k'+1} \\
&+ \sum_{k=1}^{K}\mathbf{W}_{:,:,k}\boldsymbol{x}_{t-k+1:t} + \sum_{k=1}^{K}\mathbf{V}_{:,:,k}\boldsymbol{c}_{t-k+1} + \boldsymbol{b}.
\end{aligned}
\tag{35}
$$

$\square$

The last row of equation 35 is exactly the definition of a standard conditioned temporal convolution from equation 5. The row before is a bilinear combination of $\boldsymbol{c}_{1:T}$ and $\boldsymbol{x}_{1:T}$. As a consequence of Lemma 2, conditioned (linear) hyper-convolutions are therefore a strict generalization of the standard conditioned temporal convolutions from equation 5. Note that any non-linear $\mathcal{H}^{(\mathbf{W})}$ and $\mathcal{H}^{(\boldsymbol{b})}$ that is capable of learning a linear transformation is therefore also a strict generalization of equation 5.

**Example:** $K = 1, C_{\text{out}} = 1$.

We illustrate the case for $K = 1$, i.e., the convolutions break down to a simple fully connected layer. To simplify notation, we also restrict this example to a single output channel $C_{\text{out}} = 1$.

With linear hyper-convoltions and $C_{\text{out}} = 1$, we have

$$
\mathcal{H}^{(\mathbf{W})}(\boldsymbol{c}_t) = \boldsymbol{U}\boldsymbol{c}_t + \boldsymbol{w}, \qquad \mathcal{H}^{(\boldsymbol{b})}(\boldsymbol{c}_t) = \boldsymbol{v}^T\boldsymbol{c}_t + b
\tag{36}
$$

for a weight matrix $\boldsymbol{U} \in \mathbb{R}^{C_{\text{in}} \times C_{\text{cond}}}$ and a bias vector $\boldsymbol{w} \in \mathbb{R}^{C_{\text{in}}}$ as parameters of $\mathcal{H}^{(\mathbf{W})}$ as well as weights $\boldsymbol{v} \in \mathbb{R}^{C_{\text{cond}}}$ and bias $b \in \mathbb{R}$ as parameters for $\mathcal{H}^{(\boldsymbol{b})}$. Inserting this into equation 29 leads to

$$
\begin{aligned}
z_t &= \mathcal{H}^{(\mathbf{W})}(\boldsymbol{c}_t)\boldsymbol{x}_t + \mathcal{H}^{(\boldsymbol{b})}(\boldsymbol{c}_t) \\
&= (\boldsymbol{U}\boldsymbol{c}_t + \boldsymbol{w})^T\boldsymbol{x}_t + \boldsymbol{v}^T\boldsymbol{c}_t + b \\
&= \boldsymbol{x}_t^T\boldsymbol{U}\boldsymbol{c}_t + \boldsymbol{w}^T\boldsymbol{x}_t + \boldsymbol{v}^T\boldsymbol{c}_t + b.
\end{aligned}
\tag{37}
$$

Compared to the standard formulation from equation 5,

$$
z_t = \boldsymbol{w}^T\boldsymbol{x}_t + \boldsymbol{v}^T\boldsymbol{c}_t + b,
\tag{38}
$$

equation 37 adds the bilinear term $\boldsymbol{x}_t^T\boldsymbol{U}\boldsymbol{c}_t$ while all other terms are still remaining.

## A.4 SIGNAL PROCESSING BASELINE

Sound produced in a scene arrives at the left and right ear at offset times due to the marginal difference in their distance from the source of sound (Wightman & Kistler, 1992). The coronal (back/front) asymmetry of the outer ear (pinnae) further transforms the incoming sound wave differently depending on the direction of the source (Asano et al., 1990; Cheng & Wakefield, 2001). Room effects such reverberation influence auditory localization as well (Shinn-Cunningham et al., 2005). These binaural disparities allow listeners to localize sources of sound in three dimensions (Begault et al., 2000) and gain a more complete sense of the state of the space around them.

Traditionally different effects that influence propagation between the source and the listener, i.e., different components of the mapping function between the input mono signal and output binaural signals, are addressed separately. The components are assumed to be linear time-invariant (LTI) systems and therefore completely characterized by their impulse responses. They are then combined to produce the output signals with a series of convolution operations (Savioja et al., 1999; Zotkin et al., 2004; Sunder et al., 2015; Zhang et al., 2017). A short overview follows.

**Source.** People are not omnidirectional sound sources and, unlike a loudspeaker whose spatial directivity depends only on the frequency, the human directivity pattern may depend on speech content and pose as well. Though some studies of speech directivity in controlled settings exist (Kocon &

Monson, 2018; Bellows & Leishman, 2019), in practical applications it is either ignored or approximated with a simple pattern such as a cardioid.

**Environment: room acoustics.** The reverberation is caused by the interaction of the sound field with the surrounding environment. Different approaches to room acoustic modeling exist but they are mainly divided in two groups (Välimäki et al., 2012): (1) physically accurate but computationally expensive wave-based methods that, given detailed geometry and material information, numerically solve the wave equation; and (2) methods based on geometrical acoustics (Savioja & Svensson, 2015) that ignore the wave nature of sound and assume a ray-like behaviour, and are therefore more suitable for real-time operation (wave phenomena such as diffraction is usually modeled separately (Rungta et al., 2018)). For the real-world rooms the room impulse responses (RIRs) are either measured or computed using simplified geometric models informed by some estimated room parameters. It is important to note that measurement procedure is infeasible for fully dynamic scenarios since RIRs depend on both source and listener spatial positions. The length of the RIR filters depends on the reverberation time of the environment, i.e., the time it takes for sound to decay by 60dB, which can go from less than half a second for typical office spaces to a couple of seconds for auditoriums and concert halls, and around ten seconds for large cathedrals.

**Environment: background noise.** Even in absence of other interfering sound sources, there is always some degree of ambient noise present in the environment. Usually this noise is assumed to be diffuse and independent from the listener position. It is also often assumed to be stationary and it can be estimated from short silence intervals.

**Listener.** The human body, most notably pinna, head and torso, modify the incoming acoustic waves in a way that is crucial for the spatial perception of sound. Traditionally the head-related transfer function (HRTF) is used to model these effects (Cheng & Wakefield, 2001). In theory, the HRTF is personalized to the individual (listener) and depends on the source position relative to the listener. However, most practical implementations use a generic (not-personalized) HRTF though HRTF penalization is an active research area (Bilinski et al., 2014; Yamamoto & Igarashi, 2017; Guezenoc & Seguier, 2018). Moreover, measuring HRTF in a volume is impractical and it is usually measured on a fixed radius (Li & Peissig, 2020), making the HRTF a function of the source direction only. In alternative, the boundary element method (BEM) (Katz, 2001) or finite-difference time-domain (FDTD) method (Prepeliță et al., 2016) can be used for numerical HRTF simulation using head and torso scans. Depending on the dataset, the HRTF filters are usually $2.5 - 20$ms long after removing the initial onset delays.

**Equipment.** While above components are enough to model the physics involved in the mapping between the input and the output signals, in a practical setting the capture/reproduction equipment plays a role as well. To compensate for the frequency response of the equipment and the signal-processing chain, the equalization filter should be applied. It is often assumed that this filter does not change in time and it can be estimated from a test capture.

Note that each of above steps introduces some degree of estimation, measurement or modeling errors that accumulate down the pipeline and, not being formulated in an end-to-end fashion, the solution is sub-optimal from the perspective of the particular application. Furthermore, a study showed that even using binaural room impulse responses measured for the test subject inside the test environment, the perceptual authenticity between virtual and real sound sources was not fully achieved for a dynamic scenario that allowed natural head movements of the listeners (Brinkmann et al., 2017).

Our implementation, used as the DSP baseline, computes the output binaural signals $y^{(l/r)}(t)$, for left and right ear respectively, from the input mono signal $x(t)$, $t$ being the sample index, as follows:

$$y^{(l/r)}(t) = h_{\text{eq}}^{(l/r)}(t) * h_{\text{hrtf}}^{(l/r)}(t, \boldsymbol{\theta}_t^{(\text{src,lstn})}) * h_{\text{rir}}(t, \boldsymbol{\theta}_t^{(\text{src})}, \boldsymbol{p}_t^{(\text{src})}, \boldsymbol{p}_t^{(\text{lstn})}) * x(t) + w^{(l/r)}(t), \qquad (39)$$

where

- $x(t)$ is assumed to be a clean input signal; $*$ indicates the convolution operation;
- $h_{\text{rir}}(t, \boldsymbol{\theta}_t^{(\text{src})}, \boldsymbol{p}_t^{(\text{src})}, \boldsymbol{p}_t^{(\text{lstn})})$ is the RIR computed using the image source method (Allen & Berkley, 1979) assuming a simple rectangular room and reverberation time 0.2s; $\boldsymbol{p}_t^{(\text{src})}$ and

$p_t^{(\text{lstn})}$ are the source and listener positions, and $\theta_t^{(\text{src})}$ indicates the source orientation used to simulate cardioid directivity;

- $h_{\text{hrtf}}^{(l/r)}(t, \theta_t^{(\text{src,lstn})})$ is the head related impulse response (HRTF in the time-domain) for the left / right ear; the HRTF of a KEMAR mannequin, measured in an anechoic chamber at 9600 unique discrete positions on a sphere of radius 2m is used; $\theta_t^{(\text{src,lstn})}$ indicates the direction on which the source is found with respect to the listener's front;

- $h_{\text{eq}}^{(l/r)}(t)$ is the equalization filter for the given channel, estimated from a test capture in presence of the source signal; and

- $w^{(l/r)}(t)$ is the random noise added to the given channel, generated with a power spectral density estimated from a test capture during a silence period.

Since the filters depend on source and listener positions, which in a dynamic scenario are continuously changing, the computation is done on a frame-by-frame basis using the overlap-add method, with frame length of 1024 samples and 75% overlap (at sampling rate of 48kHz).

## A.5 DATASET DESCRIPTION

**Dataset Overview.** We recorded eight different subjects, four male and four female, in an acoustically treated room. The capture contains unidirectional conversational speech, i.e., we asked participants to talk to a mannequin for 15 minutes each while walking around. We collected approximately 2 hours of mono-to-binaural audio data in total. Source and listener head positions are tracked and synchronized with the recorded audio. We use the last two minutes from each subject and a separately recorded validation sequence as test data and kept the remaining data as training data. To the best of our knowledge, this is the first binaural data capture of its kind, i.e., modeling moving trajectories between receiver and transmitter position and recorded in a regular room rather than an anechoic chamber.

**Data Capture Details.** The acoustic head and torso simulator is the GRAS KEMAR mannequin with the size large anthropometric pinnae inserts. Participants were free to walk around a 1.5m radius circle around the KEMAR mannequin, and prompted to cover as much area as a normal social conversation would. The KEMAR mannequin was wearing a B&K 4101B binaural microphone headset. The subjects wore a DPA 4060-OC microphone taped next to their mouth to capture their speech. The participants wore a modified bicycle helmet with reflective markers for head-pose tracking using an OptiTrack system. Although the KEMAR mannequin did not move, KEMAR wore a headband with reflective markers for head-pose tracking for complete source/listener head-pose tracking. All tracking information was captured with a field array of 24 OptiTrack Prime 17W cameras. The audio data is recorded at 48kHz sampling rate and rigid body tracking data is collected at 120fps via motion capture software, Motive. LTC signal is used to synchronize the audio recordings with OptiTrack data. The capture layout is schematically illustrated in Figure 7.

## A.6 EXTENDED EVALUATION

**Additional Perceptual Evaluation.** In order to back the results of the perceptual study in Table 4, we performed a side-by-side evaluation of our system and the DSP baseline, which was ranked to be the strongest competitor to our approach. In this study, participants were presented an audio snipped rendered with DSP and the same snippet rendered with our approach. The snippets are presented side-by-side in random order to ensure an unbiased evaluation. Participants were then asked to decide which of the two methods is preferable in terms of cleanliness, spatialization, and realism. We additionally gave participants the option to select *can not tell the difference* as an answer. Overall, 30 participants evaluated 360 binaural snippets generated with DSP and our method, respectively. The results in Table 7 support our findings from Table 4: our approach is preferred in terms of cleanliness and realism. For spatialization, most participants could not find a clear favorite, which is consistent with the mean opinion score of 3.75 vs. 3.76 that is reported in Table 4.

**Unseen Subjects.** While previous evaluations were based on unseen audio data from speakers that are part of the training set, we evaluate the performance on unseen speakers here. We train our model in a leave-one-speaker-out setup, i.e., we train eight models, each with another speaker being held out. Table 8 shows that our approach still outperforms the DSP baseline by a large margin

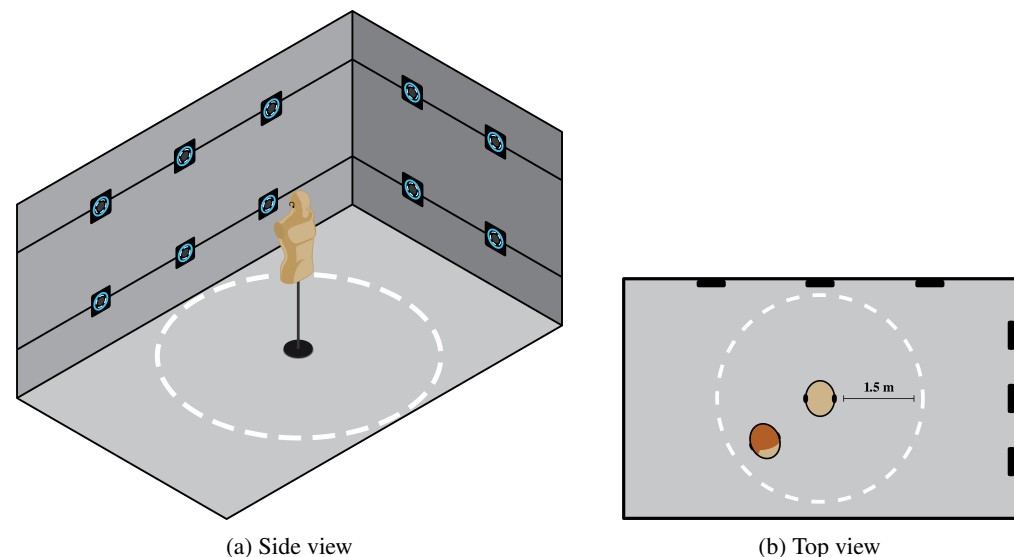

(a) Side view            (b) Top view

Figure 7: (a) Side view of capture layout. (b) Top view of capture layout. A participant moves around a KEMAR mannequin within the boundaries of a marked circle. The participant speech is recorded with a head mounted microphone and the binaural audio is captured with binaural microphones on the ears of the mannequin. Mannequin and participant positions are tracked with OptiTrack cameras mounted on the walls of the room.

Table 7: Side-by-side study of DSP vs. our system. Participants were presented two clips, one generated with DSP, one with our approach, and were then asked to tell which one they prefer.

|  | DSP preferred | ours preferred | can not tell the difference |
| --- | --- | --- | --- |
| cleanliness | 1.8% | 88.9% | 9.3% |
| spatialization | 25.9% | 27.7% | 46.4% |
| realism | 31.5% | 53.7% | 14.8% |

on unseen subjects. This is remarkable, considering that only seven different subjects are seen during training. With a more diverse dataset, we expect the generalization quality of our approach to increase significantly.

**Activation Functions.** In recent works (Sitzmann et al., 2020; Tancik et al., 2020), sine activations have been found to preserve high frequency information better than other commonly used activation functions if weights are initialized appropriately. As reconstructing high frequencies is particularly important for audio modeling, we adopt this strategy in our network. Table 9 shows a comparison of our network with ReLUs, gated convolutions as used in Van Den Oord et al. (2016), and sine activations as used in Sitzmann et al. (2020). ReLUs do not perform well on audio data: their sparse outputs are not well suited to model the smooth and sinusoid nature of waveforms. For this reason, WaveNet originally used gated convolutions, which we also find to work better than ReLUs in our task. Overall, however, we still find sine activations to produce the best results.

**Qualitative Results.** We show qualitative results on the raw waveform in Figure 8. Note the consid-

Table 8: Generalization to unseen subjects. In a leave-on-subject-out setup, our approach still outperforms the DSP baseline by a significant margin.

|  | raw waveform ($\ell_2$ error $\times 10^3$) | power spectrum ($\ell_2$ error) | phase spectrum (angular error) |
| --- | --- | --- | --- |
| DSP | 0.485 | 0.058 | 1.388 |
| ours (unseen subjects) | 0.265 | 0.058 | 1.099 |

Table 9: Effect of the activation function used in the temporal hyper-convolutions.

|  | raw waveform ($\ell_2$ error $\times 10^3$) | power spectrum ($\ell_2$ error) | phase spectrum (angular error) |
|---|---|---|---|
| ours with ReLU | 0.183 | 0.051 | 0.847 |
| ours with gated convolutions | 0.179 | 0.053 | 0.819 |
| ours with sine activation | **0.167** | **0.048** | **0.807** |

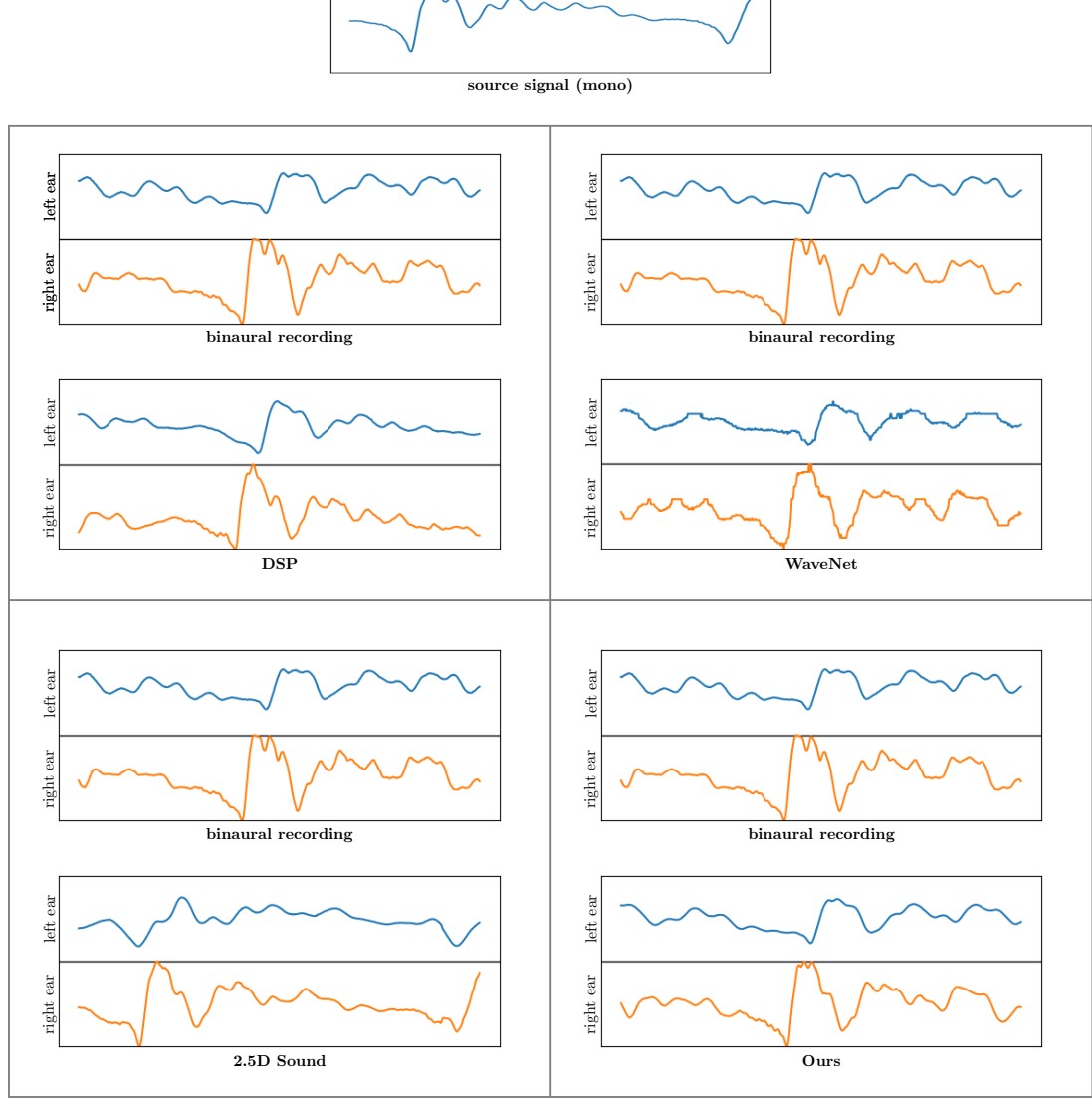

Figure 8: Qualitative results on the raw waveform. Note that 2.5D visual sound – besides having an overall inaccurate waveform reconstruction – fails to get an accurate alignment to the binaural recording. Compared to all state of the art, our approach matches the real binaural recordings best.

erable temporal shift between the mono signal captured at the source's microphone and the binaural recording. A strong binauralization system is required to accurately model not only this temporal shift but also produce a metrically correct waveform, i.e., match the shape of the binaural recording's waveform. When comparing the results for WaveNet and the 2.5D Sound architecture, it is apparent that both approaches lack in their ability to accurately match the recording's waveform.

Additionally, the 2.5D sound approach fails to align its output to the recordings. This comes of no surprise as the model is inherently designed to solve an upmixing problem, i.e., a problem where temporal shifts do not exist. Also note that due to $\mu$-law quantization that is typically applied in WaveNet, its results are non-smooth and introduce high-frequency noise due to quantization bins being misclassified. The DSP approach, which is to day the de-facto state of the art, performs more favorably in terms of temporal alignment and overall matching of the recording's waveform. Compared to our approach, however, there are significant inaccuracies – an observation that is consistent with our evaluation in Section 3.3.

