# OpenReview forum: "Neural Synthesis of Binaural Speech From Mono Audio"
_ICLR.cc/2021/Conference — ICLR 2021 Oral_

### Official Review · AnonReviewer3 · 2020-10-28
**Model, data, and evaluations for binaural audio generation from single-channel audio**

**Rating:** 7
**Confidence:** 4

**Review:**

This paper presents a neural network-based model to generate binaural audio given a single-channel audio and positions of source/listener & their angles.  The authors developed a dataset of binaural audio, which will be released on acceptance.

Technical details and model architecture are available in the body of the paper, whereas additional details such as baseline DSP-based approach, proof, and dataset are available in the appendix. The model was evaluated using the dataset developed in this work.  A demo video demonstrating the capability of the model is also provided as a supplementary material.

There are a few parts need to be addressed. (1) it is unclear why DTW-based warping is required.  IIRC the warpfield here can represent not only a shift but also other monotonic & causal such as repeating.  If there is only delay between left and right, just having a shift is enough isn't it?  It is great if the authors can explain the motivation to use warpfield more clearly.  (2) The use of hyperconvolution is an interesting idea.  The equation 5 uses conditional temporal convolution.  However, audio generative models such as WaveNet uses a different architecture; gated convolution.  The gating mechanism can give additional non-linearity and so I'm wondering if you can evaluate the performance of hyperconvolution against gated convolution.  (3) too large confidence intervals in Table 4.  Although there were many evaluations, the confidence intervals were pretty large and there were overlaps among them (e.g., small overlap between DSP and "ours" in cleanliness, large overlaps between spatialization and realism between DSP and ours).  With this result it is difficult to claim that there was a significant improvement over the baseline system.  Please check your results and design the experiment more carefully to figure out whether there is any significant difference between them.
 Conducting side-by-side comparision is one possiblity.

Comments:
-  This paper claims that it works in real time but no information about speed such as real-time factor & hardware specification are provided.
- Sampling rate information is not explicitly provided in the experiment section.
- 0.6 MOS difference is large, not "a bit".
- Modern WaveNet models often use mixture-of-logistics (refer to Parallel WaveNet paper for details) as output rather than mu-law to achieve better quality.

---

> ### Author Response · Authors · 2020-11-18
> **Thank you for the valuable comments. We will include your suggestions in a revised version.**
>
> We appreciate your detailed review and will clarify your concerns and include your suggestions in the paper.
>
> **Warping Motivation: Isn't a shift enough? Motivate more clearly.**
> Note that the source is moving, so the shift is not constant. For a simple example, consider (a) the source is in front of the listener; then the shift for left/right ear is equal, (b) source is to the left of listener; then there is a significant difference between right/left shift. Also, consider the physical effects of  a moving source: if you move towards or away from the listener, you increase or decrease the frequencies (doppler effect). From a physical point of view, all causal and monotone transformations are therefore possible. We will clarify this in the paper.
>
> **Use of gated convolutions**
> We use a sine activation as non-linearity after each (hyper-) convolution. We also experimented with gated convolutions as used in the original Wavenet but the results are worse than with sine activations. We will include a comparison in the revised version of the paper.
>
> **Perceptual Study: High standard deviations, doubts about statistical significance**
> A t-test shows that all results with the exception of "ours-vs-dsp spatialization" are statistically significant. We will mention this in the revised paper. We agree that a side-by-side perceptual study will be helpful to back our findings. We started this study and will add it to the paper as soon as it is finished.
>
> **Provide RTF and hardware specs**
> We will add an RTF and latency evaluation to the revised paper to address this point. In short: in an offline setting (mono audio is fully available prior to processing), our model has a RTF of 0.07 (i.e., 1 second of audio can be generated in 0.07 seconds). In streaming mode, our model has a latency of 32ms. All experiments are performed on a NVidia Tesla V100.
>
> **Mixture of logistics instead of mu-law for WaveNet?**
> We initially experimented with mixture-of-logistics. However, the results were of similar quality as the mu-law quantized model. Since we found mixture of logistics to be harder to optimize (longer training time until convergence) and to require a much more careful selection of hyper-parameters, we stuck to the mu-law quantization as proposed in the original, "vanilla" WaveNet.
>
> We will upload the revised version of the paper towards end of this week.

---

> ### Author Response · Authors · 2020-11-23
> **Uploaded Revised Paper**
>
> We uploaded a revised version of the paper. The major changes related to your review are
>
> **Warping Motivation** We clarified the motivation for neural time warping in Section 2.1 (second paragraph).
>
> **Use of gated convolutions** We added a comparison of sine activations to ReLUs and gated convolutions in Appendix A6 (Table 8). Overall, sine activations yield better performance than ReLUs or gated convolutions.
>
> **Perceptual side-by-side study** As suggested in your review, we conducted a perceptual side-by-side comparison between our approach and DSP (Appendix A5, Table 7). The study supports the claims from Table 4, i.e. our approach is preferred in terms of cleanliness and realism and performs equally good to DSP in terms of spatialization.
>
> **RTF and hardward specs** We added a runtime analysis in Section 3.3 (Table 5) and provide RTF and latency of our model when run on an NVidia Tesla V100.

---

> ### Comment · AnonReviewer3 · 2020-11-25
> **Thanks for the revision**
>
> Thanks for your answers and revision.  My questions were all addressed.  Glad to see its performance especially latency and RTF.

---

### Official Review · AnonReviewer1 · 2020-10-29
**An interesting paper with novel ideas and strong results**

**Rating:** 9
**Confidence:** 5

**Review:**

Strengths:
1. The paper is well written. It includes clear math notations and figures. Readers can easily follow the thought process of the authors. For example, Figure 2 shows the relation of l2 loss and phase loss with respect to target energy, indicating the importance of penalizing phase loss in the end to end system. The same observation is supported by Figure 3.

2. Strong results. The proposed end2end model significantly outperforms previous SOTA in terms of objective measures and subject tests. The video demo is very convincing. The model improved spatialization and sound quality.

3. High novelty. This paper proposes to impose monotonicity and causality to the learned warping function, which incorporates the physics of sound propagation. I am excited to another example of applying domain knowledge to an end-to-end model. The model includes two novel components: the neural warp network compensates the errors from geometry warp, the temporal convolution works as a post processing module to account for reverberation and other effects. Ablation study shows both components are critical.


To be improved:
1. The caption for Figure 4(a) seems to be incomplete.

2. It would be good to include a table to compare the proposed model with baselines in terms of model size and inference speed.

---

> ### Author Response · Authors · 2020-11-18
> **Thanks for the positive feedback**
>
> We appreciate the positive feedback for our work and are happy to include the suggested changes into the revised paper. For suggestion (2), we will include a table with model size, real-time-factor for offline inference (i.e., when the mono signal is fully available at the start of processing), and latency (i.e. when the model operates in streaming mode).
> In short, we can generate audio with a real-time-factor of 0.07 (1 second of audio takes 0.07 seconds to be binauralized) and in a streaming setup, our model runs with a latency of 32ms.
>
> We will upload the revised version of the paper end of this week.

---

> ### Author Response · Authors · 2020-11-23
> **Uploaded Revised Paper**
>
> We uploaded a revised version of the paper.
> In reply to your request for inference speed and model size, we added a runtime analysis (Table 5) to Section 3.3.

---

### Official Review · AnonReviewer4 · 2020-10-29
**Review for "Neural synthesis of binaural audio"**

**Rating:** 7
**Confidence:** 5

**Review:**

The paper is about a method for synthesizing binaural audio from a mono recording of a single speaker's speech.

First, I think the title is too general. The paper does not attempt to convert all possible sounds, but it tries to convert a single speaker's monaural speech signal to binaural audio where the speaker is moving. I think this inherent assumption is important since the method will probably not work for multiple overlapping audio sources. I suggest changing the title to "Neural synthesis of binaural speech of a single moving speaker."

The first part of the network "neural time warping" is an interesting component that is capable of adjusting the delays conditioned on the location and orientations of the source and microphone such that a location dependent binaural audio is formed by estimating time-varying delays of the original mono recording separately for two channels. It is believable that such a module would be helpful for a single moving speaker. However, such a model would not help or work when there are more than two active audio sources. A separation module would be required for that scenario. Neural time warping is an autoregressive model which can work online.

The second stage convolutional network which uses conditioned hyper convolutions is also an interesting architecture that takes the warped signals and applies time-convolutions with kernels obtained from the conditioning input which has the time-varying locations and orientations of the source and the microphone.

The section about the loss function is also interesting in that, the time domain l2 loss is shown to not work well for accurate phase estimation, so the authors propose to add a separate phase loss term to compensate for that. I think it would be better if Figure 2 is replaced with a plot of epsilon/|yhat| versus amplitude error divided by |yhat| in (a) and versus the phase error in (b). It could be clearer than the current 2D figure which is hard to interpret.

The use of "sine activation" is not well justified. "sine" activation is useful in the first layer of a "signal representation network" which is different from a signal prediction network. I do not see how and why that could be helpful here.

In terms of comparisons, 2.5D method uses visual information as conditional information to generate complex masks to produce binaural audio. In this paper, visual information is replaced with the spatial conditioning information. It would help to get more information about the window size and hop size used in 2.5D since they may be an important factor that relates to the amount of delays they can introduce.  For wavenet comparison, it was not clear how the wavenet model was trained to generate binaural data. Did it use any conditioning information? If so , how? Was it applied in an auto-regressive way with randomized sampling? The wavenet audio example sounded noisy which is not typical of wavenet generated audio. It looks like the DSP method can utilize a listener specific HRTF which may be difficult to incorporate for the proposed neural model. Is it an important issue?

How does the model generalize to unseen speakers and rooms? The training and testing strategy uses the same room and the same speaker(s).  Would we have any problem when the monaural audio is recorded in some other room with some other speaker?

In Figure 8, maybe it is OK not to draw the original binaural signal for every method.

In general, I liked the neural warping and conditional convolutional components which are interesting and I liked the analysis about the loss function. The approach is an interesting way to obtain binaural version of a monaural mono speaker recording in a room. The dataset produced for the paper would also be useful for research.

**Update after revision**

The revision improved the paper. Thanks for taking care of my comments.

Justification of sine activations, generalization to unseen speakers experiment are nice additions.

The new title is a bit better and I think it may be OK since the goal is to perform a moving source simulation for single speech sources. Multiple speech sources can be simulated separately and added together as mentioned.
The authors may consider possibly a better name: "Neural binaural synthesis from mono speech" which emphasizes that the synthesized target is "binaural speech" from a single speech recording.

Just a few more points.

1. I think it is essential in wavenet to apply the model in an auto-regressive fashion over samples. Just using the network architecture and the loss function from wavenet is not equivalent to "using a wavenet model" since an essential part of the model is the autoregressive sampling which makes sure the samples are dependent and coherent. Without auto-regressive sampling, the resulting sound is poor as observed by the authors. So, I suggest to emphasize that "autoregressive sampling" is not performed in the paper to avoid misleading the readers.

2. More explanation of 2.5D is appropriate. One wonders if using a larger STFT window size would improve its results.

---

> ### Author Response · Authors · 2020-11-18
> **Thanks for the careful review, we are happy to include your suggestions**
>
> **Title**
> We agree that the title is quite generic. We suggest to change the title to "Neural Synthesis of Binaural Speech" or "Neural Synthesis of Binaural Speech from Mono Audio". Since basically all spatialization papers assume that the mono source contains the signal from a single speaker, we would argue that this does not need to be contained in the title specifically.
>
> **Time warping would not work for two sound sources in mono input**
> That is true. Existing binauralization approaches assume a single-source mono signal, so this is not a restriction over existing work. Note that due to audio being additive, we can easily generalize to multiple speakers by running our system for each speaker individually and then summing the result -- this would assume that each speaker has his own mono input audio. In this work, we do not address source separation, which is a challenging and unsolved research problem on its own.
>
> **A different representation of Fig. 2 could be better**
> While a classical 1D plot would be easier to read, dividing by |yhat| (i.e., normalizing over the target energy) hides the impact of the target energy on the amplitude loss (Fig. 2a). We believe it is important to see this impact as it gives insights into the behavior of the l2 loss among signal components of different energy.
>
> **Sine activation needs better justification**
> We build on recent work on [1,2] that provides an empirical and theoretical analysis why a sine activation function filters better high frequency components in a MLP structure. We compared with ReLu, and the results, if we follow the proposed initialization in [1],  were better. We will add a further comparison to GLUs as used in WaveNet to the revised paper. Since it is not the main contribution of the paper, we leave further analysis for the future.
>
> **2.5D sound: more information**
> We use a window size of 1600 samples and a hop length of 480 samples (10ms) for the STFT. Therefore, modeling delays <10ms requires non-trivial manipulation of the phase information in the complex spectrogram, which is a more difficult operation than modeling delays in time-domain. We will add this information to the paper.
>
> **WaveNet comparison needs clarifications**
> The WaveNet comparison uses the source/listener positions as conditioning (i.e., we replace the linguistic/acoustic features that are used in the original paper). Conditioning follows section 2.5 in [3], i.e., layer input and conditioning features (source/listener positions) go through a 1D convolution each, mapping them to the same number of output channels, and are then added. While we used the WaveNet architecture, we did not apply the network in an autoregressive way because - in contrast to speech synthesis from text - our input is not the left-shifted output signal but the mono signal, which is known in advance and therefore does not require autoregressive forwarding. The noisy output is caused by two factors: (1) we have 48kHz audio, which is more difficult to model than 16-24kHz which is usually used by TTS systems. [4] show that the quality of WaveNet degrades with higher sampling rates. (2) WaveNet has to spend a considerable amount of capacity on modeling the non-linear source-to-listener time shift between the mono and binaural signal and, in consequence, struggles more to generate truly clean audio.
>
> **Listener-specific HRTFs in DSP**
> Some studies suggest that the impact of individualized HRTFs is low or negligible [5,6]. Either way, even for DSP, individual HRTFs need to be measured for the respective listener, so it does not yield a benefit over our system which could be trained or fine-tuned on listener-specific data.
>
> **Generalizability to unseen speakers**
> For the case of unseen speakers, we will add an experiment with left-out speakers to the paper to verify a reasonably good generalization. For rooms, our model can not generalize to heavy reverberation such as a cathedral if this is not included in the training data. For typical office/living-room sized rooms, our model produces satisfying results. Recall that DSP based approaches also use approximations for rooms such as a shoe-box model.
>
> We will upload a revised paper including clarifications and your suggestions towards end of the week.
>
> [1] Implicit Neural Representations with Periodic Activation Functions, Sitzmann et.al.
>
> [2] Fourier Features Let Networks Learn High Frequency Functions in Low Dimensional Domains, Tancik et.al.
>
> [3] Wavenet: A generative model for raw audio, van den Oord et.al, 2016
>
> [4] Deep Voice: Real-time Neural Text-to-Speech, Arik et.al., 2017
>
> [5] Do we need individual head-related transfer functions for vertical localization? The case study of a spectral notch distance metric, Geronazzo et.al.
>
> [6] Generic HRTFs may be good enough in virtual reality. Improving source localization through cross-modal plasticity, Berger et.al., 2018

---

> ### Author Response · Authors · 2020-11-23
> **Uploaded Revised Paper**
>
> We uploaded a revised version of the paper. The major changes with respect to your request are
>
> **Title** Updated to Neural Synthesis of Binaural Speech.
>
> **Sine activations need better justification** While an evaluation of activation functions is not a contribution of the paper and left for future analysis, we added a comparison of sine activations with ReLUs and gated convolutions in Appendix A6 to provide more empirical justification for the choice of sine activations.
>
> **2.5D sound: more information** We added more details about the 2.5D sound model in Section 3.3.
>
> **WaveNet comparison needs clarification** We added the clarification to Section 3.3.
>
> **Generalization to unseen speakers** We added a leave-one-speaker-out experiment to Section 3.3 (Table 6) and show that on unseen speakers our model still performs better than the DSP baseline.

---

### Decision · Program_Chairs · 2021-01-07
**Final Decision**

**Decision:**

Accept (Oral)

**Comment:**

+ Interesting method for binaural synthesis from moving mono-audio
+ Nice insight into why l2 isn't the best loss for binaural reconstructions.
+ Interesting architectural choice with nice results.
+ Nicely motivated and clearly presented idea -- especially after addressing the reviewers comments.

I agree with the idea of a title change. While I think its implied that the source is probably single source, making it explicit would make it clearer for those not working in a closely related topic. Hence, "Neural Synthesis of Binaural Speech from Mono Audio" as suggested in the review process sounds quite reasonable.